# Common Variants in the *TYR* Gene with Unclear Pathogenicity as the Cause of Oculocutaneous Albinism in a Cohort of Russian Patients

**DOI:** 10.3390/biomedicines12102234

**Published:** 2024-10-01

**Authors:** Olga Shchagina, Anna Stepanova, Polina Mishakova, Vitaliy Kadyshev, Nina Demina, Ludmila Bessonova, Sofya Ionova, Daria Guseva, Andrey Marakhonov, Rena Zinchenko, Sergey Kutsev, Aleksander Polyakov

**Affiliations:** Research Centre for Medical Genetics, Moscow 115522, Russia

**Keywords:** oculocutaneous albinism, *TYR*, hypomorphic variants

## Abstract

**Background:** oculocutaneous albinism (OCA) is a hereditary impairment of skin, hair, and eye pigmentation. The most common form of albinism is autosomal recessive albinism, caused by mutations in the *TYR* gene, accounting for approximately 40–50% of all cases of the disease in European populations. Common hypomorphic variants in the *TYR* gene could lead to a mild form of albinism in a compound heterozygous state with a pathogenic variant. **Methods**: we examined by allele specific MLPA a cohort consisting of 118 unrelated patients with albinism and 10 parents of these patients. The control cohort consisted of 200 unexamined Russian residents. **Results**: the patients with albinism were divided into three groups: without pathogenic variants in the *TYR* gene—70 patients, with one pathogenic variant in the *TYR* gene—20 patients, and with two pathogenic variants in the *TYR* gene—28 patients. Among the 20 patients with a single heterozygous variant in the *TYR* gene, 15 patients had the c.575C>A p.(Ser192Tyr) variant, and 15 had the c.1205G>A p.(Arg402Gln) variant. Both the c.575C>A p.(Ser192Tyr) and c.1205G>A p.(Arg402Gln) variants were identified in 12 patients. In addition to the aforementioned variants, an intronic variant c.1185-6208A>G (rs147546939) was identified in seven patients. **Conclusions**: the frequencies and the number of alleles c.575A, c.1205A, and c.1185-6208G in different groups of patients and the control group were compared. In this study, we demonstrate that the complex alleles [c.575C>A p.(Ser192Tyr); c.1205G>A p.(Arg402Gln)] and [c.575C>A p.(Ser192Tyr); c.1185-6208A>G; c.1205G>A p.(Arg402Gln)] are associated with oculocutaneous albinism, which is consistent with findings from other researchers.

## 1. Introduction

Oculocutaneous albinism (OCA) is a hereditary impairment of skin, hair, and eye pigmentation, also characterized by a combination of ophthalmological symptoms. OCA is caused by pathogenic variants in genes that determine the production of melanin. To date, there are eight known genes that are causative of OCA with autosomal recessive inheritance: *TYR* (OCAIA and OCAIB), *OCA2* (OCAII and Albinism, brown oculocutaneous), *TYRP1* (OCAIII), *SLC45A2* (OCAIV), *OCA5* (OCAV), *SLC24A5* (OCAVI and variations in skin pigmentation), *LRMDA* (OCAVII), and *DCT* (OCAVIII). Furthermore, pigmentation disorders may be associated with variants in the *GPR143* gene, leading to an X-linked recessive type of disease. Pigmentation, variations in skin color, refers to limited human traits where the phenotypic and genotypic variance is much greater between populations than within them [1]. Pigmentation impairment or ocular manifestations characteristic of albinism can also be observed in patients with syndromal pathology, such as mutations in the *PAX6* gene [2].

The most common form of albinism is autosomal recessive albinism, caused by mutations in the *TYR* gene, accounting for approximately 40–50% of all cases of the disease in European populations [3,4,5,6]. Multiple studies have shown that a significant proportion of patients with albinism only have a singular pathogenic variant in the *TYR* gene [2,3,6,7,8,9,10]. In some, but not all, of these patients, pathogenic causal variants in other genes have been identified using genome/exome analysis methods [5].

Since the pigmentation of the eyes, hair, and skin represents a continuous spectrum of phenotypes, researchers have considered the presence of common hypomorphic variants in the *TYR* gene that could lead to a mild form of albinism in a compound heterozygous state with a pathogenic variant, or possibly even in a homozygous state [7]. Two common missense substitutions have been described in the *TYR* gene: rs1042602 c.575C>A p.(Ser192Tyr) and rs1126809 c.1205G>A p.(Arg402Gln), which are also frequently observed as parts of a complex allele. It is suggested that the cis-position of these variants is a result of multiple recombination events and de novo mutations in intron 3 of the *TYR* gene [8]. Functional studies have shown that the c.1205G>A p.(Arg402Gln) variant leads to a 75% reduction in tyrosinase catalytic activity compared to the wild type [9,10,11], while the c.575C>A p.(Ser192Tyr) variant results in a 40% reduction in the enzyme’s activity [12]. However, there was no proof of the causality of these variants with respect to albinism either individually or as part of a complex allele in trans-position with pathogenic variants [13], although many studies have demonstrated a non-random distribution of these variants in patients with albinism [7,14]. The frequency of the complex allele [c.575C>A p.(Ser192Tyr); c.1205G>A p.(Arg402Gln)] is significantly higher in patients with albinism heterozygous for a pathogenic variant in the TYR gene or in patients without a pathogenic *TYR* variant. However, the 1.9% frequency of the complex allele among non-Finnish Europeans is also too high for the variant to be considered pathogenic [9].

Haplotype analysis suggests that the allele [c.575C>A p.(Ser192Tyr); c.1205G>A p.(Arg402Gln)] originated due to recombination, and that several [c.575C>A p.(Ser192Tyr); c.1205G>A p.(Arg402Gln)] haplotypes differ among individuals affected by oculocutaneous albinism and among those in the general population [8,15]. There has been a demonstrated correlation between light/dark skin pigmentation in South Asian people and the polymorphic variant c.575C>A p.(Ser192Tyr) [16]. In Europe, it has also been shown that the presence of this allele is associated with eye color, the presence of freckles, and skin pigmentation [17].

The aim of the current study was to evaluate the contribution of the c.575C>A p.(Ser192Tyr) and c.1205G>A p.(Arg402Gln) variants to the genetic structure of albinism in a cohort of Russian patients. The study also included the rs147546939 c.1185-6208A>G variant, located in intron 3 of the *TYR* gene, which has been associated with albinism in European patients as part of the haplotype [c.575C>A p.(Ser192Tyr); c.1185-6208A>G; c.1205G>A p.(Arg402Gln)] (rs1042602A–rs147546939G–rs1126809A) [18].

## 2. Materials and Methods

We examined a cohort consisting of 118 unrelated patients with albinism and 10 parents of these patients. Direct automated Sanger sequencing of the *TYR* gene was conducted for 44 patients using primers that flank the coding sequences of all of the exons in the gene: TYR1F0:TTCCTGCAGACCTTGTGAGGAC; TYR1F2:CCTTCCGTCTTTTATAATAGGACC; TYR1R1:CAGTTGAATCCCATGAAGTTGCC; TYR1R2:GGTTCCTCCCTACTCTGACATC; TYR2F:ACTGGTGGTGACAATTTGTTTAAC; TYR2R:TCCTAGGACTTTGGATAAGAGAC; TYR3F:GGATAATCACATAGGTTTTCAGTC; TYR3R:TCCAATGAGCACGTTATTTATAAAG; TYR4F:AATATGTTTCTTAGTCTGAATAACCT; TYR4R: AACACTAGATTCAGCAATTCCTCT; TYR5F:AGGTGTCTACTCCAAAGGACTGT; TYR5R:TGCAAATGGTCTTTACAGAAAAATAC. In 74 cases, the analysis was conducted on an Illumina Miseq sequencer using a miseq reagent kit v2 (500-cycles). The probes were prepared using ultramultiplex PCR followed by sequencing (AmpliSeq™, San Diego, CA, USA). The analysis was conducted using a custom «Albinism» panel, which included coding sequences of the following genes: *C10orf11*, *TYR*, *SLC24A5*, *OCA2*, *MC1R*, *MITF*, *SLC45A2*, *TYRP1*, and *GPR143*. Samples of 20 patients with a singular heterozygous variant in the TYR gene underwent quantitative analysis using SALSA MLPA Probemix P325 OCA2 (MRC Holland, The Netherlands). The control cohort consisted of 200 unexamined Russian residents. The c.575C>A p.(Ser192Tyr), c.1205G>A p.(Arg402Gln), and c.1185-6208A>G variants were analyzed using allele-specific probe ligation with subsequent PCR using the following oligonucleotide probes: MTYR192FN:GTTCGTACGTGAATCGCGGTACGTTGATGCACTGCTTGGGGGATC; MTYR192FM:GTTCGTACGTGAATCGCGGTACGTTTTGGATGCACTGCTTGGGGGATA; MTYR192R:TGAAATCTGGAGAGACATTGATTTTGCCGATGCGATCCGATGCCTTCATG; MRS147546939FN:GTTCGTACGTGAATCGCGGTACAACGAATCAGTTCCCAAATATCCAACTA; MRS147546939FM:GTTCGTACGTGAATCGCGGTACGAATCAGTTCCCAAATATCCAACTG; MRS147546939R:TCTTAAGGCTTCCTTGTTTGTTCTTTTTTTAAGATGCGATCCGATGCCTTCATG; MTYR402FN:GTTCGTACGTGAATCGCGGTACAGTATTTTTGAGCAGTGGCTCCG; MTYR402FM:GTTCGTACGTGAATCGCGGTACTGCAGTATTTTTGAGCAGTGGCTCCA; MTYR402R:AAGGCACCGTCCTCTTCAAGAAGATGCGATCCGATGCCTTCATG. The results were visualized with PAAG electrophoresis (Figure 1).

It should be noted that our study was limited to investigating associations of c.575C>A p.(Ser192Tyr), c.1205G>A p.(Arg402Gln), and c.1185-6208A>G variants with TYR gene-related albinism. We have not investigated the effect of variants on other genetic variants of albinism.

The allele frequency statistical analysis for Control cohort and different Patient cohorts was based on the χ^2^ test for a 2 × 2 contingency table with the alleles divided into two groups: the associated allele and all the others. The level of significance was determined by a distribution with one degree of freedom. Additionally, we applied the relative risk criterion (odds ratio—OR). Most of statistical analysis was performed using GraphPad Prism 8.0.1 for Windows (GraphPad Software, San Diego, CA, USA). *p*-value less than 0.05 is considered to be statistically significant.

## 3. Results

As a result of the diagnostic DNA studies, pathogenic and likely pathogenic variants in the *TYR* gene were identified in 48 unrelated patients. Biallelic variants were detected in 28 patients, while monoallelic variants in the *TYR* gene were identified in 20 patients. The genotypes of the patients are presented in Appendix A.

Among the 20 patients with a single heterozygous variant in the *TYR* gene, 15 patients had the c.575C>A p.(Ser192Tyr) variant (three in a homozygous state), and 15 had the c.1205G>A p.(Arg402Gln) variant (seven in a homozygous state). Both the c.575C>A p.(Ser192Tyr) and c.1205G>A p.(Arg402Gln) variants were identified in 12 patients. In addition to the aforementioned variants, a search was conducted for the intronic variant c.1185-6208A>G (rs147546939), which has been suggested to be associated with albinism in European patients [18]. The intronic variant was identified in seven patients.

Family analysis conducted in five cases with two exonic variants showed that the c.1185-6208A>G variant was identified as part of the complex haplotype [c.575C>A p.(Ser192Tyr); c.1185-6208A>G; c.1205G>A p.(Arg402Gln)] in trans-position with a described pathogenic variant in four of them. Additionally, in one family, a haplotype without the intronic variant [c.575C>A p.(Ser192Tyr); c.1205G>A p.(Arg402Gln)] in trans-position with a pathogenic variant was identified.

To assess the correlation between the albinism phenotype and the c.575C>A p.(Ser192Tyr); c.1185-6208A>G; c.1205G>A p.(Arg402Gln) variants, we genotyped the entire group of 118 patients diagnosed with oculocutaneous albinism and a Control cohort of 200 unexamined Russian residents.

The patients with albinism were divided into three groups: without pathogenic variants in the *TYR* gene—70 patients, with one pathogenic variant in the *TYR* gene—20 patients, and with two pathogenic variants in the *TYR* gene—28 patients. The results are presented in Table 1.

In the control cohort, no deviations from the Hardy–Weinberg equilibrium were detected for all three markers. When comparing the entire patient group with the control cohort, a significant association with the albinism phenotype was found only for the c.1205A allele of the c.1205G>A p.Arg402Gln variant (χ^2^ = 27.728, p(χ^2^) < 0.001, OR 2.489 (1.772–3.494)). Comparing the control cohort with the subgroup of patients without pathogenic variants in the *TYR* gene also revealed significant differences in the allele frequency of the c.1205G>A p.(Arg402Gln) variant, which was significantly lower in the control cohort (0.24) compared to patients without *TYR* gene mutations (0.35) (p(χ^2^) = 0.011).

Allele frequencies also showed significant differences between patients with two pathogenic variants and the control group. For the c.575C>A variant, the frequency of the c.575A allele (p(χ^2^) = 0.008) was significantly lower than in the control cohort, while for the c.1205G>A variant, the frequency of the c.1205A allele was significantly higher (p(χ^2^) < 0.0001). However, unlike in patients without mutations or with a single mutation, the allele frequencies were lower than in the control group. The c.575C>A p.(Ser192Tyr) and c.1205G>A p.(Arg402Gln) substitutions were also significantly more common in patients with two pathogenic *TYR* variants than in patients without mutations (p(χ^2^) = 0.016; 0.003). These results suggest that one or several pathogenic variants are presented as part of a complex allele. Indeed, in six patients, homozygous for the common pathogenic variant c.650G>A (p.Arg217Gln), the c.1205G>A p.(Arg402Gln) variant was also identified in a homozygous state, indicating that the disease in these patients is caused by the complex allele [c.650G>A (p.Arg217Gln); c.1205G>A p.(Arg402Gln)]. Additionally, the homozygous c.1205G>A p.(Arg402Gln) variant was found in a patient with the homozygous pathogenic c.1044+1G>T variant, three compound heterozygous patients with c.650G>A;1037-7T>A variants, and singular compound heterozygous patients with the c.650G>A;896G>A, c.650G>A;1204C>T, c.650G>A;c.1037G>A, and c.1279G>T;302G>A variants. Aside from that, this variant was found in a heterozygous state in patients with albinism caused by c.650G>A;1037-3C>G, c.650G>A;1037-7T>A, c.1A>G;230G>A variants. With that, none of the patients with a pathogenic TYR variant within the complex allele containing the c.1205G>A p.(Arg402Gln) variant had the c.575C>A p.(Ser192Tyr) variant.

When comparing the group of patients with a single *TYR* gene variant to the control cohort, significant differences were found for all three studied variants (Table 2).

Significant differences at all three investigated points—c.575C>A p.(Ser192Tyr); c.1185-6208A>G; c.1205G>A p.(Arg402Gln)—were noted when comparing groups of patients without mutations and patients with a single pathogenic variant in the *TYR* gene (p(χ^2^) −0.011; <0.001; 0.022, respectively)—the allele frequencies were significantly higher in the subgroup of heterozygous patients.

Following the comparison of allele frequencies, the number of alleles c.575A, c.1185-6208G, and c.1205A in the genotype was determined for the variants c.575C>A p.(Ser192Tyr); c.1185-6208A>G; c.1205G>A p.(Arg402Gln) in the control cohort, in all patients with albinism, as well as in groups of patients with two pathogenic variants, with one pathogenic variant, and without pathogenic variants (Table 3).

In the control cohort, carriers of at least one of the alleles (c.575A, c.1185-6208G, or c.1205A) are significantly less frequent than among patients with oculocutaneous albinism. It was also shown that the accumulation of one, three, and four alleles is significantly more common in patients with a single described pathogenic variant in the *TYR* gene compared to the control group, whereas two variants are significantly more common in patients with two pathogenic variants (Figure 2).

## 4. Discussion

The three examined alleles were most frequently found in the subgroup of patients with a single pathogenic variant, followed by patients with four alleles. This can be explained by the presence of a complex allele comprising three variants or the presence of a fourth common variant (within another complex allele with a pathogenic variant) that does not influence disease development. Indeed, in this group of patients, a combination of alleles, c.575A, c.1185-6208G, and c.1205A, was identified in four out of five cases. This combination was identified as a complex allele during family analysis in four cases. In patients with four variants, alleles c.575A, c.1185-6208G, or c.1205A, were present in the genotype in all five cases. As the fourth variant, c.1205A was identified in four cases and c.575A—in one case.

The obtained data allows us to suggest two hypotheses:

**H1.** 
*The c.1205G>A p.(Arg402Gln) variant is encountered as part of a complex allele with different pathogenic variants, as well as with the c.575C>A p.(Ser192Tyr) variant. This suggests that because it is prevalent in the population, different mutational events occur on chromosomes with this variant, as well as on chromosomes without it.*


**H2.** 
*The complex allele [c.575C>A p.(Ser192Tyr); c.1205G>A p.(Arg402Gln)], and even more so [c.575C>A p.(Ser192Tyr); c.1185-6208A>G; c.1205G>A p.(Arg402Gln)], is either a hypomorphic allele in a compound heterozygous state with a pathogenic variant, or a marker haplotype, on which a relatively recent mutational event in the regulatory regions of the TYR gene has occurred, leading to albinism.*


If the hypothesis about the marker haplotype is correct, the search for other mutational events should not cease. Recently, the potential role of the *TYR* c.-301C>T (rs4547091) variant, which is located in the gene’s promoter region, in the development of oculocutaneous albinism was demonstrated [19]. A significant reduction in promoter activity in vitro was shown [20]. Due to the relatively high frequency of this variant in the population, its role in the development of albinism was studied as part of the TYR c.[-301C;575A;1205A] haplotype. It was shown that *TYR* c.-301C>T [rs4547091] modulates the penetrance of the common missense variant, *TYR* c.1205G>A (p.Arg402Gln) [rs1126809]. The T allele has a protective effect in individuals with the c.1205G>A substitution (OR < 0.7), while the C allele in the cis-position with c.1205G>A leads to a high risk of albinism (OR > 24). Homozygosity for the haplotype comprising three variants, *TYR* c.[-301C;575C>A;1205G>A], is associated with a high likelihood of an albinism diagnosis (OR > 82). According to the RuExac database [21], the c.-301C>T (rs4547091) variant has an allele frequency of 0.00069, indicating that the frequency of the allele-reducing *TYR* activity is 0.99931. The allele frequency of the c.1205G>A (p.Arg402Gln) variant is 0.220790378; it was found in 1120 individuals. The T allele, which does not influence or enhance *TYR* function, was identified in three individuals out of 2910 on four chromosomes, but only one individual with a homozygous c.-301C>T variant also had the c.1205G>A(p.Arg402Gln) variant. Therefore, in 1120 individuals, 168 of whom are homozygous for the c.1205G>A(p.Arg402Gln) variant, this variant is part of the *TYR* c.[-301C;1205G>A] complex allele, including 168 homozygotes. This does not allow for the consideration of this variant as the cause of albinism in Russian patients. Therefore, its analysis was not included in the current study.

Thus, in this study we demonstrate that the complex alleles [c.575C>A p.(Ser192Tyr); c.1205G>A p.(Arg402Gln)] and [c.575C>A p.(Ser192Tyr); c.1185-6208A>G; c.1205G>A p.(Arg402Gln)] are associated with oculocutaneous albinism, which is consistent with findings from other researchers [7,14].

The candidate variant c.1185-6208A>G (rs147546939) was identified during the analysis of TYR gene haplotypes in a cohort of Danish patients with oculocutaneous albinism and in a specific subgroup of this cohort, which had only one pathogenic variant in the TYR gene in intron 3. The minor allele (G) frequency was significantly higher in the albinism cohort compared to both the Control group from the Genome Denmark study (*p* < 0.0001) and the gnomAD database. The c.1185-6208G allele was found only in patients with albinism who were heterozygous for TYR gene mutations, always in trans-position with the pathogenic variant, or in patients without pathogenic variants. It was shown that in patients with a single pathogenic TYR gene variant, the haplotype of the second chromosome included the following variants: c.575C>A p.(Ser192Tyr); c.1185-6208A>G; c.1205G>A p.(Arg402Gln). Analysis of intron 3 sequence surrounding the c.1185-6208A>G variant indicated the potential for the formation of two pseudoexons, 25 bp and 104 bp in length, respectively. However, functional studies were unable to confirm this because of the large size of intron 3. Our study also revealed a significant difference in allele frequencies of the intronic variant c.1185-6208A>G between Russian albinism patients and the control cohort.

The regulation of skin pigmentation is a complex multi-level process. The main genes, including the TYR considered in this work, are involved in the processes of tyrosine metabolism. It is interesting to note that in addition to albinism, there are other conditions, such as vitiligo [11,22,23], in the development of which genes of the same metabolic pathway are involved. Unfortunately, today it is extremely difficult to draw a line that would make it possible to distinguish between normal population variants of these genes and disease-causing variants [1,16,17].

## 5. Conclusions

The modifying effect of cis-positioned regulatory variants on disease penetrance has been previously demonstrated for several disease groups [24]. Based on the accumulated data, further investigation is required to identify variants in non-coding regions of the *TYR* gene, which are part of the haplotype with the common *TYR* gene variants: c.575C>A p.(Ser192Tyr); c.1185-6208A>G; c.1205G>A. These variants could potentially be the missing link explaining the phenotype manifestation. 

## Figures and Tables

**Figure 1 biomedicines-12-02234-f001:**
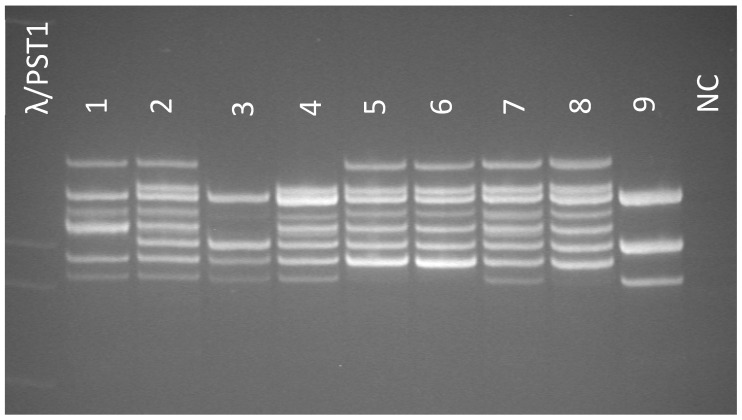
Results of visualization for the c.575C>A p.(Ser192Tyr), c.1205G>A p.(Arg402Gln), and c.1185-6208A>G variants. λ/PST1—molecular weight marker—phage’s λ DNA treated with PST1 restriction endonuclease, Lines 1–9—results of MLPA analysis for samples with different genotypes, NC—negative control.

**Figure 2 biomedicines-12-02234-f002:**
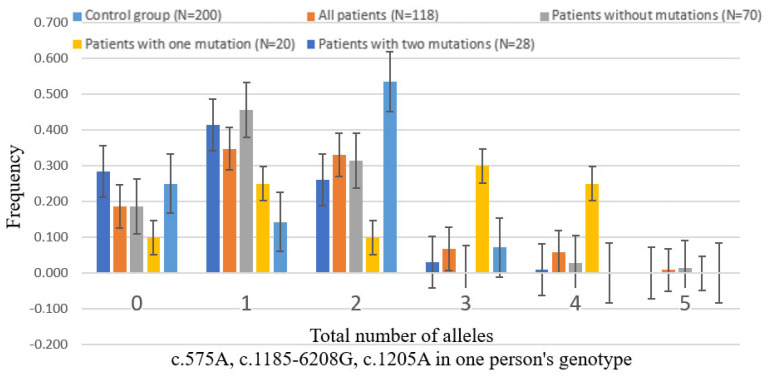
Accumulation number of alleles c.575A, c.1185-6208G, and c.1205A in the studied groups.

**Table 1 biomedicines-12-02234-t001:** Genotyping results of patient groups with albinism and the control group.

Examined Group (Number of People)	Number of Chromosomes	c.575C>A p.Ser192Tyr		c.1185-6208A>G		c.1205G>A p.Arg420Gln	
		Number of alleles (allelic frequency)					
Allele		c	a	a	g	g	a
Control cohort (200)	400	291 (0.73)	109 (0.27)	391 (0.98)	9 (0.02)	304 (0.76)	96 (0.24)
All patients (118)	236	175 (0.74)	61 (0.26)	226 (0.96)	10 (0.04)	132 (0.56)	104 (0.44) *
Subgroups of patients							
Patients without *TYR* variants (70)	140	103 (0.74)	37 (0.26)	137 (0.98)	3 (0.02)	91 (0.65)	49 (0.35) *
Patients with one *TYR* variant (20)	40	22 (0.55) *	18 (0.45) *	33 (0.83) *	7 (0.17) *	18 (0.45) *	22 (0.55) *
Patients with two *TYR* variants (28)	56	50 (0.89) *	6 (0.11) *	56 (1.0)	0 (0)	23 (0.41) *	33 (0.59) *

* Values that vary significantly between the Patient group and the Control cohort.

**Table 2 biomedicines-12-02234-t002:** Comparing the group of patients with a single *TYR* gene variant to the control cohort.

Variant	Allele	Control	*TYR* Hetero	χ^2^	p(χ^2^)	OR (CI)
**Number of Chromosomes**		**400**	**40**			
c.575C>A p.Ser192Tyr	c	291	22	5.58	0.018	2.189 (1.143–4.193)
	a	109	18			
c.1185-6208A>G	a	391	33	24.134	<0.001*	9.226 (3.802–22.389)
	g	9	7			
c.1205G>A p.Arg420Gln	g	304	18	17.806	<0.001*	3.838 (2.055–7.167)
	a	96	22			

* Values that vary significantly between the patient group and the control cohort.

**Table 3 biomedicines-12-02234-t003:** The number of alleles c.575A, c.1185-6208G, and c.1205A in the Control cohort, in all patients with albinism, as well as in groups of patients with two pathogenic variants, with one pathogenic variant, and without pathogenic variants.

Total Number of c.575A, c.1185-6208G, c.1205A Alleles in One Person	0	1	2	3	4	5
Control cohort (200 people)	57	83	52	6	2	0
All patients (118 people)	22	41	39	8	7	1
χ^2^	3.861 *	1.423	1.806	2.519	6.565 *	-
p(χ^2^)	0.049	0.231	0.179	0.113	0.01	-
Patients without mutations (70 people)	13	32	22	0	2	1
χ^2^	2.662	0.377	0.768	-	1.225	-
p(χ^2^)	0.103	0.534	0.385	-	0.266	-
Patients with one mutation (20 people)	2	5	2	6	5	0
χ^2^	3.171	2.063	2.513	25.702 *	33.996 *	nd
p(χ^2^)	0.075	0.151	0.113	<0.0001	<0.0001	nd
Patients with two mutations (28 people)	7	4	15	2	0	0
χ^2^	0.149	7.709 *	8.998 *	1.245	-	-
p(χ^2^)	0.699	0.006	0.003	0.262	-	-

* Values that vary significantly.

## Data Availability

The data presented in this study are available on request from the corresponding author.

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
