# Peer review of "Common Variants in the TYR Gene with Unclear Pathogenicity as the Cause of Oculocutaneous Albinism in a Cohort of Russian Patients"

_biomedicines, 2024, doi:10.3390/biomedicines12102234_

Round 1

Reviewer 1 Report

Comments and Suggestions for Authors

Authors have examined a cohort consisting of 118 unrelated patients with albinism and 10 parents of these patients. The control cohort consisted of 200 unexamined Russian residents, which means that an acceptable sample size is enrolled comparing the previous studies and prevalence of oculocutaneous albinism. Then authors have divided the patients with albinism into three groups: without pathogenic variants in the TYR gene – 70 patients, with one pathogenic variant in the TYR gene – 20 patients, and with two pathogenic variants in the TYR gene – 28 patients. According to the authors, among the 20 patients with a single heterozygous variant in the TYR gene, 15 patients had the c.575C>A 21p.(Ser192Tyr) variant, and 15 had the c.1205G>A p.(Arg402Gln) variant. Both the c.575C>A p.(Ser192Tyr) and c.1205G>A p.(Arg402Gln) variants were identified in 12 patients. Furthermore,, an intronic variant c.1185-6208A>G (rs147546939) was identified in seven patients. The frequencies and the number of alleles c.575A, c.1205A and c.1185-6208G in different groups of patients and the control group were compared. This paper demonstrate that the complex alleles [c.575C>A p.(Ser192Tyr); c.1205G>A p.(Arg402Gln)] and [c.575C>A 27p.(Ser192Tyr); c.1185-6208A>G; c.1205G>A p.(Arg402Gln)] are associated with oculocutaneous albinism, which is consistent with findings from other researchers. Literature review and introduction is acceptable, methods are well-described and replicable, results are clearly presented and nicely discussed. Also, authors have suggested their suggestions for future research and direction. Overall, this paper adds to the literature and enhances our understanding of oculocutaneous albinism. The novelty of this work is that during the analysis of intron 3 sequence surrounding the c.1185-6208A>G 261 variant indicated the potential for the formation of two pseudoexons, 25 bp and 104 bp in length, respectively. However, functional studies were unable to confirm this because of the large size of intron 3. This study also shows a significant difference in allele frequencies of the intronic variant c.1185-6208A>G between Russian albinism patients and the control cohort.

I failed to see where authors have mentioned the limitation(s) of their study and suggest they include the limitations(s), if any.

Author Response

Thank you very much for taking the time to review this manuscript. Please find the detailed responses below. 

Thank you for your careful and attentive review of our manuscript.

Comment 1: 

"I failed to see where authors have mentioned the limitation(s) of their study and suggest they include the limitations(s), if any."

Response 1:

In accordance with your comment, limitation have been added to the materials and methods section (corrections highlighted blue).

Reviewer 2 Report

Comments and Suggestions for Authors

This present study examined the contribution of variants to the genetic structure of albinism in a cohort of patients and compared that with a control group who do not have this condition. The manuscript is well written and adds to the current knowledgebase on this topic. In the study, the authors demonstrated that the complex alleles are associated with oculocutaneous albinism, and provided a thorough discussion on two hypotheses based on their findings. The figures and tables in the results section are well organised and presented, but the methods section is very brief and they did not explain the statistical analyses. Based on their findings, the authors recommended further investigation to identify variants in non-coding regions of the TYR gene, and suggested that the variants could potentially be “the missing link” to explain the phenotype manifestation.

Author Response

Thank you very much for taking the time to review this manuscript. Please find the detailed responses below.

Thank you for your careful work with the manuscript. We are glad that you liked our research and the manuscript. 

Comments 1:

"The figures and tables in the results section are well organised and presented, but the methods section is very brief and they did not explain the statistical analyses. "

Response 1:

In accordance with your comment, we have added information about statistical analysis methods (highlighted yellow) to the materials and methods section.

Reviewer 3 Report

Comments and Suggestions for Authors

The study is nicely designed. However, a much more detailed discussion of the achieved results can be provided.

The authors performed a thorough genetic study on oculocutaneous albinism using a group of 318 patients. The authors show that Ser192Tyr and Arg402Gln mutations of TYR gene are associated with oculocutaneous albinism. The study is nicely designed and meet the standards of modern human genetics.

However, the manuscript is quite short in my opinion. Some minor points should be properly discussed. I recommend minor revision and address the following suggestions:

-        The introduction section is missing information on the metabolic relation between TYR, Tyr, dopachrome and melanin. The authors should provide a short passage on this topic. For example, see [1].

-        The impaired pterin metabolism as an etiology factor for vitiligo may probably have similar origin to the TYR gene-related albinism since in both cases impaired tyrosine metabolism is observed. In recent years, much progress has been developed in this field by the groups of Schallreuter [2,3] and Telegina [4,5].

-        P. 7, line 275: “The findings and their implications should be discussed in the broadest context possible.” So why the authors do not even try make a discussion of the established findings in this manuscript? The discussion can be improved.

1.          Liu, F.; Wen, B.; Kayser, M. Colorful DNA polymorphisms in humans. Semin. Cell Dev. Biol. 2013, 24, 562–575, doi:https://doi.org/10.1016/j.semcdb.2013.03.013.

2.          Wood, J.M.; Chavan, B.; Hafeez, I.; Schallreuter, K.U. Regulation of tyrosinase by tetrahydropteridines and H2O2. Biochem. Biophys. Res. Commun. 2004, 325, 1412–1417, doi:https://doi.org/10.1016/j.bbrc.2004.10.185.

3.          Schallreuter, K.U.; Chavan, B.; Rokos, H.; Hibberts, N.; Panske, A.; Wood, J.M. Decreased phenylalanine uptake and turnover in patients with vitiligo. Mol. Genet. Metab. 2005, 86 Suppl 1, S27-33, doi:10.1016/j.ymgme.2005.07.023.

4.          Telegina, T.A.; Lyudnikova, T.A.; Buglak, A.A.; Vechtomova, Y.L.; Biryukov, M. V; Demin, V. V; Kritsky, M.S. Transformation of 6-tetrahydrobiopterin in aqueous solutions under UV-irradiation. J. Photochem. Photobiol. A Chem. 2018, 354, 155–162, doi:https://doi.org/10.1016/j.jphotochem.2017.07.029.

5.          Telegina, T.A.; Vechtomova, Y.L.; Borzova, V.A.; Buglak, A.A. Tetrahydrobiopterin as a Trigger for Vitiligo: Phototransformation during UV Irradiation. Int. J. Mol. Sci. 2023, 24, 13586, doi:10.3390/ijms241713586.

Author Response

Response to Reviewer 3 Comments

Thank you very much for taking the time to review this manuscript. Please find the detailed responses below.

Thank you for your positive review and careful review of our work.

We are pleased to expand our knowledge about the regulation of skin pigmentation based on the materials proposed by the reviewer.

Comments 1 -        P. 7, line 275: “The findings and their implications should be discussed in the broadest context possible.” So why the authors do not even try make a discussion of the established findings in this manuscript? The discussion can be improved.

Response 1In accordance with your recommendation, we have expanded the introduction and discussion sections (highlighted with a gray), and links have also been added to the list of references.